# Synergic Effect of Phthalide Lactones and Fluconazole and Its New Analogues as a Factor Limiting the Use of Azole Drugs against Candidiasis

**DOI:** 10.3390/antibiotics11111500

**Published:** 2022-10-28

**Authors:** Piotr Krężel, Teresa Olejniczak, Aleksandra Tołoczko, Joanna Gach, Marek Weselski, Robert Bronisz

**Affiliations:** 1Faculty of Biotechnology and Food Science, Wrocław University of Environmental and Life Sciences, Norwida 25, 50-375 Wrocław, Poland; 2Faculty of Chemistry, University of Wrocław, F. Joliot-Curie 14, 50-383 Wrocław, Poland

**Keywords:** synergic effect, FIC, MIC, in silico studies, *Candida albicans*, 3-*n*-butylphthalide, fluconazole, azoles, 1,2,3-triazole

## Abstract

The resistance of *Candida albicans* and other pathogenic yeasts to azole antifungal drugs has increased rapidly in recent years and is a significant problem in clinical therapy. The current state of pharmacological knowledge precludes the withdrawal of azole drugs, as no other active substances have yet been developed that could effectively replace them. Therefore, one of the anti-yeast strategies may be therapies that can rely on the synergistic action of natural compounds and azoles, limiting the use of azole drugs against candidiasis. Synergy assays performed in vitro were used to assess drug interactions Fractional Inhibitory Concentration Index. The synergistic effect of fluconazole (**1**) and three synthetic lactones identical to those naturally occurring in celery plants—3-*n*-butylphthalide (**2**), 3-*n*-butylidenephthalide (**3**), 3-*n*-butyl-4,5,6,7-tetrahydrophthalide (**4**)—against *Candida albicans* ATCC 10231, *C. albicans* ATCC 2091, and *C. guilliermondii* KKP 3390 was compared with the performance of the individual compounds separately. MIC_90_ (the amount of fungistatic substance (in µg/mL) inhibiting yeast growth by 90%) was determined as 5.96–6.25 µg/mL for fluconazole (**1**) and 92–150 µg/mL for lactones **2–4.** With the simultaneous administration of fluconazole (**1**) and one of the lactones **2–4**, it was found that they act synergistically, and to achieve the same effect it is sufficient to use 0.58–6.73 µg/mL fluconazole (**1**) and 1.26–20.18 µg/mL of lactones **2–4**. As fluconazole and phthalide lactones show synergy, 11 new fluconazole analogues with lower toxicity and lower inhibitory activity for CYP2C19, CYP1A2, and CYP2C9, were designed after in silico testing. The lipophilicity was also analyzed. A three-carbon alcohol with two rings was preserved. In all compounds 5–15, the 1,2,4-triazole rings were replaced with 1,2,3-triazole or tetrazole rings. The hydroxyl group was free or esterified with phenylacetic acid or thiophene-2-carboxylic acid chlorides or with adipic acid. In structures **11** and **12** the hydroxyl group was replaced with the fragment -CH_2_Cl or = CH_2_. Additionally, the difluorophenyl ring was replaced with unsubstituted phenyl. The structures of the obtained compounds were determined by ^1^H NMR, and ^13^C NMR spectroscopy. Molecular masses were established by GC-MS or elemental analysis. The MIC_50_ and MIC_90_ of all compounds **1–15** were determined against *Candida albicans* ATCC 10231, *C. albicans* ATCC 2091, AM 38/20, *C. guilliermondii* KKP 3390, and *C. zeylanoides* KKP 3528. The MIC_50_ values for the newly prepared compounds ranged from 38.45 to 260.81 µg/mL. The 90% inhibitory dose was at least twice as high. Large differences in the effect of fluconazole analogues **5–15** on individual strains were observed. A synergistic effect on three strains—*Candida albicans* ATCC 10231, *C. albicans* ATCC 2091, *C. guilliermondii* KKP 339—was observed. Fractional inhibitory concentrations FIC_50_ and FIC_90_ were tested for the most active lactone, 3-*n*-butylphthalide, and seven fluconazole analogues. The strongest synergistic effect was observed for the strain *C. albicans* ATCC 10231, FIC 0.04–0.48. The growth inhibitory amount of azole is from 25 to 55 µg/mL and from 3.13 to 25.3 µg/mL for 3-*n*-butylphthalide. Based on biological research, the influence of the structure on the fungistatic activity and the synergistic effect were determined.

## 1. Introduction

### 1.1. Candida Albicans as a Potentially Infectious Disease Agent

Candidiasis is an infection caused by both *Candida albicans* and *Candida* non-albicans (NAC) species, most often endogenously, as the primary source of infection is the gastrointestinal tract [1,2]. There is also human-to-human transmission and infection in the hospital environment when invasive medical procedures are performed. *C. albicans* and *C. glabrata* are currently the most common fungal pathogens found in humans. It is predicted that 75% of women worldwide will develop candidiasis at least once in their lifetime. In the United States, between 2013 and 2017 [3], *Candida* yeasts were responsible for a quarter of deaths caused by hospital-acquired bloodstream infections. They are responsible for disseminated candidiasis, intravascular candidiasis, and even central nervous system infections. In highly developed countries such as the USA, Norway, and the UK, an increase in mortality due to general infection of the organism with opportunistic *Candida* strains is increasingly observed [4,5]. That is why it is so important to take action to counteract this problem, which is becoming more and more serious every year and constitutes a worrying threat to public health [6].

### 1.2. Pharmacological Use of Fluconazole and Associated Risks

It is known that fluconazole is one of the most used anti-candidiasis drugs [7]. It is a compound belonging to the azole group. It has a broad spectrum of inhibition against most *Candida* yeasts, *Cryptococcus*, and other dermatophytes responsible for systemic and superficial fungal infections in humans. In pharmacology, it has been used as a first-line treatment since 1990.

Fluconazole, due to the presence of two triazoles in its structure and a phenyl ring containing two fluorine atoms in the meta position, interferes with sterol synthesis by inhibiting lanosterol 14α-demethylase, one of the critical enzymes of the cytochrome P450 system [8]. Therefore, fluconazole in therapy can lead to hepatotoxicity [9,10]. This phenomenon is a significant problem for people with reduced immunity and immunosuppression. The most vulnerable groups are people with AIDS, patients undergoing chemotherapy, and those with respiratory diseases, including severe COVID-19 [11].

### 1.3. Mechanism of Action of Fluconazole against Candida Yeast

The mechanism of action of fluconazole is based on the inhibition of the expression of the gene *ERG11*, responsible for the biosynthesis of 14a-lanosterol demethylase (CYP51A1) of cytochrome P450 involved in the conversion of lanosterol to 4,4-dimethylcholesta-8 (9), 14,14-trien-3β-ol, an important stage of biosynthesis of sterols. The inhibition of ergosterol synthesis increases the permeability of the cell membrane as it is a key component of the yeast cell membrane. This causes the accumulation of toxic 14-methyl sterols in fungal cells. Therefore, fluconazole interferes with the synthesis and growth of cell walls as well as cell adhesion, and hence it has a fungistatic effect [12,13,14,15,16] (Figure 1a).

Azole compounds used in medicine and plant protection are responsible for the resistance of *Candida* strains [17]. There are several mechanisms of resistance to azole drugs shown by *Candida* strains. The first is the introduction of multi-drug pumps into the fungal cell wall so that the drug can be pumped out of the cell and its effect is limited. The pumps arise because of point mutations of the *CDR1*/*CDR2* and *MDR1* genes, as well as the action of the transcription factors TAC1 and MDR1, which code for efflux pumps [18,19]. Another mechanism leading to drug resistance is mutations in the *ERG11* gene, which is the target of action of azole drugs. The mutations change the enzyme binding site, which prevents azole binding and lowers the drug’s effectiveness [20]. In vitro studies have shown that in *C. albicans* strains, in addition to increased gene overexpression, point mutations occur in the *ERG11* gene. Through point mutations involving the change of lysine 143 to arginine, glutamic acid 266 to aspartic acid, valine 404 to leucine, and valine 488 to isoleucine in isolated *C. albicans* strains, a conformational change in lanosterol 14α-demethylase was discovered that reduces the effective binding of fluconazole molecules, thereby effectively attenuating its effects [20]. Fungal cells may also be able to develop new pathways of cell membrane synthesis that are not interfered with by azole drugs. Therefore, the effectiveness of the drug is lowered, and the fungi retain the functionality of the cell membranes. Due to the more and more frequent acquisition of drug resistance by fungal strains, more research has been carried out on new azole drugs against which fungi are no drug-resistant [3,21,22].

### 1.4. Phthalide Lactones, Isolated Plants of the Apiaceae Lindl Family

Bicyclic phthalide lactones are produced by *Ligusticum officinale* (lovage), *Ligusticum chuanxiong*, *L. wallichii* (Chinese lovage) *Apium graveolens* (celery), and *Petroselinum crispum* (parsley). Some of them have been described in the literature [23,24,25,26]. Their structure was determined, and synthetic methods were proposed. Some biological activities were tested based on using the above plants in traditional Chinese medicine [27,28]. Phthalide lactones have fungistatic activity [29].

3-*n*-Butylphthalide (NBP) is a natural compound approved by the Chinese National Administration of Medical Products for treating ischaemic stroke [30,31,32,33,34]. Repeated studies have shown that the NBP demonstrated multidirectional neuroprotective effects in ischaemic conditions. It has also been shown to reduce brain swelling, improving microcirculation in the ischaemic area. It inhibits oxidative stress and apoptosis of nerve cells [35,36,37,38].

### 1.5. Mechanism of Action of 3-n-Butylphthalide on Candida Albicans Cells

The presence of 3-*n*-butylphthalide in the cytosol leads to an increased production of ROS (reactive oxygen species) by the mitochondria, as demonstrated by measuring the mitochondrial membrane potential (Δψm). The accumulation of reactive oxygen species leads to the direct destruction of the mitochondrial cell membrane and the release of large amounts of ROS into the cytosol. In addition to their oxidative action, it disrupts cellular homeostasis. ROS destroys DNA deposited in the cell nucleus and degrades enzymes and their production by disrupting translation and lipid peroxidation, leading to the destruction of the cell membrane. 3-*n*-Butylphthalide indirectly initiates the process of necrosis through its presence. 3-*n*-Butylphthalide influences the regulation of the *CDR1* and *CDR2* genes, contributing to the accumulation of this water-insoluble substance (logP = 2.8) [33,39,40] (Figure 1b).

The overall aim of the study was to investigate the correlation between fluconazole, newly developed derivatives, and phthalide lactones. The obtained results confirmed the purpose of the research. In most cases, phthalide lactones, combined with azole compounds, were effective in reducing the concentrations of fluconazole and its derivatives due to synergistic action in the reaction system. The developed methods, especially the quantitative proportions of individual compounds needed to achieve this effect, may be one of the likely strategies of their application against *Candida* strains having genes for resistance to fluconazole. It is worth mentioning that the use of phthalide lactones, characterized by low toxicity and a relatively small effect on cellular metabolism, to reduce the number of azoles used in pharmacy may help to inhibit the resistance trend, especially in the hospital environment.

## 2. Results and Discussion

### 2.1. Synthesis of Phthalide Lactones ***2–******4*** and Analogues of Fluconazole ***5–******15***

#### Synthesis of Phthalide Lactones **2–****4**

Phthalide lactones—3-*n*-butylphthalide (**2**), 3-*n*-butylidenephthalide (**3**), 3-*n*-butyl-4,5,6,7-tetrahydrophthalide (**4**)—were obtained in two-stage syntheses, in which the intermediate product 3-*n*-butyl-3-hydroxylactone formed from dibutylcadmium and the appropriate anhydride was subjected to reduction with NaBH_4_ reduction or dehydration with p-toluenesulfonic acid (Figure 2) [22].

In the next stage of the work, it was planned to change the fluconazole fragments and observe their effect on the biological activity and synergies (Figure 3).

A three-carbon alcohol with two rings was preserved. In all compounds **5–15**, the 1,2,4-triazole rings were replaced with 1,2,3-triazole or tetrazole rings. The hydroxyl group was free or esterified with chlorides of phenylacetic acid or 2-thiophenecarboxylic acid or adipic acid. In structures 11 and 12 in the site of the hydroxy and difluorophenyl group, the -CH_2_Cl or =CH_2_ fragment was inserted. Additionally, the difluorophenyl ring was replaced with unsubstituted phenyl. The structures of the obtained compounds were determined by ^1^HNMR and ^13^C NMR spectroscopy, and molecular mass was established by GC-MS or elemental analysis. The GC analyses performed were helpful in the synthesis, determining the purity of the compounds obtained (Figure 3).

The structures of all tested compounds are shown in Figure 4.

Fluconazole, i.e., 1,3-di(1H-1,2,4-triazol-1-yl)-2-(2,4-difluorophenyl)propan-2-ol, is a very biologically active compound. Due to the presence of two triazoles in its structure and a phenyl ring containing two fluorine atoms and a hydroxyl group, fluconazole belongs to both azoles and hetero compounds. It is also an aromatic compound. Therefore, when planning the synthesis of 1,3-di(1H-1,2,3-trizol-1-yl)-2-(phenyl)propan-2-ol (**5**), we only changed two items: the type of the triazole ring; and the removal of the heteroatoms in the phenolic ring. This allowed us to obtain a less toxic compound (see 2.2) and showed no inhibitory effect on cytochrome CP2C19 like fluconazole (see 2.2) [41,42].

1,3-di(1H-1,2,3-trizol-1-yl)-2-(phenyl)propan-2-ol (**5**) is one of the regioisomers obtained in the Grignard reaction of magnesium phenyl bromide with 1,3-dichloroacetone [43], and sodium 1,2,3-trazolate (Figure 5).

The structures of the next compounds (**6–10**) determine the extent to which the blocking of the hydroxyl group will affect the fungistatic activity. Their activity was compared to the obtained **13–15** alcohols by reaction of the sodium salt with 1,2,3-triazole and 1,3-dichloro-2-propanol in acetone, **13**, **14**, **15** (Figure 6). These compounds were separated by column chromatography using silica gel and a mixture of dichloromethane:methanol. The composition of the isolated compounds was consistent with the percentage composition from the chromatographic analysis (Figure 7).

The isolated alcohols **13–15** were substrates in the syntheses of esters **6**, **7**, **8**, **10**. The reactions were carried out in the presence of pyridine by adding the appropriate chloride of phenylacetic acid or 2-thiophenylcarboxylic acid or adipic acid. The scheme of an exemplary synthesis is shown in Figure 8.

We also synthesized three compounds with tetrazole rings [44,45]. All of them were obtained from mixtures of three regioisomers formed in the synthesis: 1-(tetrazol-1-yl)-3-(tetrazol-2-yl)propan-2-yl thiophen-2-ylcarboxylic ester (**9**), 1-[2-(2H-tetrazol-2-ylmethyl)prop-2-en-1-yl]-1H-tetrazole (**11**), 1,3-di(tetrazol-2-yl)-2-chloromethylpropane (**12**). The first is ester formed in a mixture of triazole alcohols, to which the esterification procedure was carried out without separation into individual isomers. Two other analogues of fluconazole, **11** and **12**, were obtained according to the scheme below (Figure 9 and Figure 10).

[2-(2H-tetrazol-2-ylmethyl)prop-2-en-1-yl]-1H-tetrazole (**11**) was prepared by heating tetrazole sodium salt with a solution of 3-chloro-2-(chloromethyl)prop-1-ene in acetonitrile for 4 days. 1,3-di(tetrazol-2-yl)-2-chloromethylpropane (**12**), also obtained from tetrazole sodium salt and 2-[3-chloro-2-(chloromethyl)propyl]-2H-tetrazole in acetonitrile, was stirred and heated to gentle reflux for 10 days.

The structures of all compounds (**2–15**) were determined based on ^1^HNMR and ^13^CNMR GC-MS spectroscopic analysis.

### 2.2. In Silico Studies

The data from in silico ADME analyses were included in considerations. Lipophilicity (Figure 11a), toxicity to rat cells (Figure 11b), and the inhibition of enzymes responsible for biotransformation and detoxification of drugs are all extremely important (Figure 12) [41,42,46,47,48,49].

#### 2.2.1. Phthalide Lactones **2–4**

The obtained phthalide lactones **2–4** (Figure 2), i.e., internal esters, have a similar structure and differ in the number and position of double bonds. 3-*n*-Butylphthalide (**2**) and 3-*n*-butylidenephthalide (**3**) are aromatic compounds and therefore have a flat structure. 3-Butyl-4,5,6,7-tetrahydrophtalide (**4**) has only one double bond, and the cyclohexene ring is not flat; its conformation may change depending on the environment. Lactones **2–4** have a molecular weight of 188–194 g/mol. Fluconazole has a different chemical structure. The consequence of different chemical structure is different lipophilicity. Fluconazole dissolves better in water than fats and lactones **2–4** and vice versa Lactones **2–4** and fluconazole differ in their toxicity (Figure 11b). Fluconazole is 3 to 10 times more toxic than the lactones shown. The difference in toxicity between the individual lactones is surprising. Fluconazole is a potent inhibitor of CYP2C19, responsible in mammal’s liver for detoxifying up to 20% of drugs (2 3]. 3-*n*-butylphthalide inhibits CYP1A2 and three times more toxic 3-butylidenephthalide than CYP2C9. The least poisonous 3-*n*-butyl-4,5,6,7-tetrahydrophtalide (**4**) is not a potent inhibitor of CYP.

Enzymes belonging to cytochrome P450 are a group of enzymes that have in their structure a heme molecule in common. They all play a key role in the metabolization of xenobiotics in all representatives of eukaryotes [50,51]. When analyzing cytochrome P450 in the context of mammalian cells, it should be emphasized that the largest group of xenobiotics consists of pharmaceuticals and their derivatives, which are products of intracellular (endogenous) metabolism. An analysis of the cDNA of mammalian cells confirmed that expression of these enzymes occurs in most tissues, with the highest levels reached in liver tissue. One of its primary tasks is detoxification of the body. The three families of cytochrome P450 enzymes, CYP1, CYP2, and CYP3, which are responsible for the biotransformation of a large proportion of the drugs commonly used in pharmacology, are of greatest importance in determining the pharmacokinetics and the effects of chemicals on mammalian organisms.

#### 2.2.2. Fluconazole (**2**) and Its Analogues **5–15**

The toxicity of fluconazole determined as LC_50_ = 584 mg/kg is 1/3 higher than that of its structural analogue, where the 1,2,4-triazole rings have been replaced with 1,2,3-triazole rings, and the phenolic ring has no fluoric substituents. The lipophilicity of both compounds is very similar. 1,3-di(1H-1,2,3-triazole-1-yl)-2-phenyl propan-2-ol (**5**) is not a CYP2C19 inhibitor, although fluconazole and esters **6–10** show such inhibition (Figure 12).

An interesting aspect in the perspective of toxicity and the inhibitory activity of cytochrome P450 enzymes is the obtained esters **6–10**. All esters obtained in the ADME analysis are classified as inhibitors of CYPs. The triazole alcohol esters (**6**, **7**, **8**), despite their different triazole ring configurations, inhibit only one CYP2C19 enzyme, while compound **9**, which is a tetrazole alcohol ester, inhibits CYP1A2, 2C14, and 3A4, three key enzymes from the point of view of xenobiotic metabolism. The ability to inhibit the above-described enzymes must be combined with the fact that the -OH group, which occurs freely in alcohol, is blocked (Figure 4 and Figure 12).

### 2.3. Fungistatic Activity

The results of experimentally determined fungistatic activity of fluconazole and lactones **2–4** and fluconazole analogues **5–15** against the references strains *Candida albicans* ATCC 10231*, C. albicans* ATCC 2091, *C. albicans* clinical isolate AM 38/20, *C. zeylanoides* KKP 3528, and *C. guilliermondii* KKP 3390, are shown in Figure 13 (Appendix A).

In vitro antifungal evaluation showed that phthalide lactones had lower fungistatic activity than fluconazole (Figure 13). In the case of phthalide lactones, a 50% inhibition of the growth of microorganisms was observed at concentrations in the range 30.79–85.04 µg/mL. In the case of compound **3**, having a double bond at C-3, higher activity was observed against the ATCC 2091 strain, in which 17.61 µg/mL inhibited the growth of this organism by 50%. A similar situation was observed in the KKP 3390 strain, where the concentration needed to achieve a 50% growth reduction was around 36.96 µg/mL.

The fungistatic activity of compounds with phenyl **5**, **6, 7** was determined against 5 strains of pathogenic yeasts: *Candida albicans* ATCC 10231, *C. albicans* ATCC 2091, *C. albicans* clinical isolate AM 38/20: *C. zeylanoides* KKP 3528, and *C. guilliermondii* KKP 3390. The most sensitive strain is *C albicans* ATCC 2091, for which 60% inhibition was observed at concentrations in the range 69.18–82.9 µg/mL. The other strains required at least twice this amount. An analysis of the fungistatic activity of unsubstituted alcohols **13–15** was surprising (Figure 13).

More active are alcohols where the first and third carbon atoms are bonded to triazoles in position 1 or 2. These compounds are more potent against *C. zeylanoides* KKP 3528 and *C. guilliermondii* KKP 3390 MIC_50_ 45.48–77.89 µg/mL, against *C. albicans* strains MIC_50_ 122.29 to 208.8 µg/mL. In all tests with alcohols, the dose needed to reach MIC_90_ is twice as high. These compounds have lipophilicity not recommended for bioassays, suggesting poorer bioavailability. Esterification with phenylacetic acid is favorable in the case of *C. albicans* ATCC 2091 alcohol **13**: MIC_50_ 128.43 µg/mL ester **7**: 82.90 µg/mL and alcohol **15**: MIC_50_ 170.59 ester **6**: 69.18 µg/mL, while for *Candida zeylanoides* KKP 3528 and *Candida guilliermondii* KKP 3390 opposite dependencies were observed.

Relatively low concentrations were detected for the ester with adipic acid (**10**), inhibiting the growth of tested *Candida* yeast strains from 38.45 to 133.49 µg/mL, excluding the most resistant to all tested azoles, *C. albicans* clinical isolate AM 38/20. Not confirmed hopes with esters 8 and 9 for high biological activity. (Figure 13)

#### 2.3.1. Synergic Effect of Lactones **2–4** and Fluconazole in Combination

In the next stage of the experiment, it was decided to determine the effects of fluconazole (**1**) and phthalide lactones (**2**,**3**,**4**) to study the mutual interactions between them.

For this purpose, the checkerboard protocol previously described in the literature [52] was used. Fluconazole and phthalide lactones dissolved in DMSO at a 10 μg/mL concentration were added to a 96-well microtiter plate and diluted using the double dilution method to produce mixtures of both compounds at different concentrations and ratios. A standardized inoculum of the microorganism was then placed. It was decided to select two reference strains of *C. albicans* and the strain of *C. guilliermondii* for the experiment. The plates prepared in this way were placed in an incubator until an optical density (OD) of 1.0 was reached in the reaction systems.

The absorbance values obtained allowed the determination of the minimum inhibitory concentration of specific fluconazole and lactone mixtures and the fractional inhibitory concentration (FIC). This coefficient quantifies the interaction between the compounds present in the mixture.

There are three types of interaction between biologically active compounds: synergism, neutral action, and antagonism. The synergism (∑FIC ≤ 0.5) is a state in which two existing compounds by common occurrence in the mixture mutually reinforce each other’s effect, and thus their combined effect is greater than the sum of the effects of their separate actions.

However, the most common, the presence of both does not affect the cells in the system, neutral action (0.5 < ∑FIC ≤ 4. 0). When ∑FIC > 4.0, we observed antagonism, which is a state in which two existing compounds by common occurrence in the mixture mutually weaker each other’s effect. Thus, their combined effect is less than the sum of the effects of their separate actions [53].

The results presented in Table 1 show that in all cases, fluconazole and phthalide lactones exerted a synergistic effect in combination, thus lowering the concentration of each compound in the culture mixture. In the case of compound **1**, the concentration to achieve a 90% reduction in ATCC 109231 growth was about 45x lower, and in the case of fluconazole, about 2× lower. A similar phenomenon can be observed with compound **3**, whose concentration was 22.25–1.20 μg/mL, sufficient to obtain MIC_90_. A similar situation is observed in the case of compound **4**, where the reduction of the amount of lactone needed for significant inhibition of *Candida* was several dozen times lower than in case of the use of lactones without the addition of fluconazole. It should also be noted that the activity of fluconazole in all cases increased with the presence of phthalide lactones. Thus, a concentration several times lower than when fluconazole alone was used was needed to achieve the desired effect.

Due to the use of 3-*n*-butylphthalide (**2**) for treatment in China, it was assumed that if any pharmaceutical company decides to use the synergistic effect in the treatment of candidiasis, there is a good chance that they will choose this compound.

The synergistic effect occurring between 3-butylidenophthalide and fluconazole was interesting (cf. Table 1). The mechanism of action of fluconazole is based on the inhibition of the expression of the gene ERG11, responsible for the biosynthesis of ergosterol. Inhibition of ergosterol synthesis increases the permeability of the cell membrane as it is a key component of the yeast cell membrane, and hence it has a fungistatic effect [12,13,14,15,16]. There are several mechanisms of resistance to azole drugs shown by *Candida* strains. One of them is the introduction of multi-drug pumps: the CDR1/CDR2 and MDR1 into the fungal cell wall so that the drug can be pumped out of the cell and its effect is limited [18,19].

The fungistatic effect of 3-*n*-buthylphthalide (**2**) and its analogues 3 and 4 are related to the accumulation of reactive oxygen species (ROS). ROS leads to the direct destruction of the mitochondrial cell membrane and the release of large amounts of ROS into the cytosol, destroys DNA deposited in the cell nucleus, and degrades enzymes and their production by disrupting translation and lipid peroxidation, leading to the destruction of the cell membrane. 3-*n*-Butylphthalide influences the regulation of the CDR1 and CDR2 genes, contributing to the accumulation of this water-insoluble substance [33,39,40].

The synergistic mechanism between lactones 2–4 and azoles [39,40] is based on promoting drug absorption and inhibiting drug efflux by downregulating the transcription of drug transporters CDR1 and CDR2. In all cases studied, the use of this compound with fluconazole resulted in an enhanced synergistic effect. Nevertheless, after ADME prediction, this compound showed the highest toxicity (1865 mg/kg). Analyzing the structure of the phthalide lactones studied reveals a correlation between the spatial structure of the compounds and the increase in their toxicity. Compound **4**, which does not have a conjugated double bond system in the A ring in space, adopts a *cis* conformation. In addition, the aliphatic fragments C-8, C-9, C-10, and C-11 can move freely in space, which significantly facilitates the hydroxylation reaction of this compound by the cell. A similar situation regarding the aliphatic fragment occurs in the case of compound **2**, but its rings in space will be flat due to the occurrence of resonance in ring A. Analyzing the structure of compound **4**, attention should be paid to the double bond at C-3, which significantly stabilizes the structure of the aliphatic fragment. Compound **3** in space will assume the flattest form in comparison with **2** and **4**, which could translate into a hindrance to the biotransformation process.

#### 2.3.2. Synergistic Effect of 3-Butylophthalide (**2**) and Azole Derivatives (**5**, **6**, **8**, **9**, **10**, **12**, **13**) against Selected *Candida* Strains

In the final stage of the research, an interrelationship between 3-*n*-butylphthalide and fluconazole analogues **5–10**, **12**, **13** were studied.

The results from the experiment confirmed that in almost all cases, the mixture consisting of an azole derivative and 3-*n*-butylphthalide (**2**) reached up to six times lower concentration values with a simultaneous higher inhibition coefficient. The interaction of 3-*n*-butylphthalide with the derivatives showed a synergistic effect in all cases.

The research hypothesis assumes that the fungistatic action of 3-*n*-butylphthalide is enhanced in combination with relatively few toxic derivatives of fluconazole (Figure 14). In the case of *C. albicans* ATCC 109231, the presence of 3-*n*-butylphthalide in the reaction mixture in the range 0.25–6.25 μg/mL reduced more than six-fold the concentration of all tested azole derivatives (**5, 6, 8, 9, 10, 12, 13**) needed to stop yeast cell proliferation by 50 and 90%. An analogous situation can be observed in the results of studies conducted for *C. albicans* strain ATCC 2091, where 3-*n*-butylphthalide concentration around 3.35 μg/mL was sufficient to reduce the concentration of the derivatives 10-fold. An interesting phenomenon from the biological point of view is the lack of homogeneity of the results between the tested strains. Each of them had a different tolerance to azole derivatives, which directly reflected on the obtained results of synergistic action with 3-*n*-butylphthalide.

A similar situation is observed in the case of 3-*n*-butylphthalide, which, even within species, exerts different degrees of inhibition on yeast. This is mainly due to the origin of the specific strains. Yeast as unicellular organisms, though admittedly to a lesser extent than bacteria, have the ability to rapidly adapt and change metabolic pathways, which can cause the phenomenon where each strain reacts differently to a specific reaction environment. Nevertheless, it is worth noting the general trend of 3-*n*-butylphthalide and azole compounds in the mixture (Figure 14), which indicates that the use of 3-*n*-butylphthalide with less active fluconazole derivatives compensate for their activity so that a mixture of the two could theoretically replace fluconazole.

It should be noted that despite the achievement of the research objective and the positive results, the in vitro study at the microbiological level was primarily based on in silico-calculated toxicity analyses based on the likelihood of obtaining chemical structures for existing compounds. An exception to this rule is 3-*n*-butylphthalide, which has been comprehensively determined pharmacokinetically in animal models and clinical trials [33]. Therefore, the considerable concern is that the effects of modified fluconazole derivatives may be completely different in the in vivo test. As mentioned in the introduction, the resistance mechanisms to azole drugs are different [12,13,14,15,16,17,18,19,20,21,22]; despite these reservations, we tested two fluconazole-resistant strains isolated from patients (Figure 15a,b). We hope that in medical research, the obtained results may be valuable data for potential fungistatic factor design strategies.

We also performed tests with two fluoro-resistant strains of *C. albicans*, AM 38/22 and AM 38/680. The first strain inhibited yeast growth by 90% at concentrations above 256 μg/mL, and the second strain was above 1024 μg/mL. We did a few repetitions. In Figure 15a,b, we present the degree of growth inhibition of *C. albicans* AM38/22 and AM38/680 against 3-*n*-butylphthalide (**2**), fluconazole (**1**), as well as azoles **5–9**, **11–15** in concentrations of 300 μg/mL and 150 μg/mL. The determined degree of growth inhibition was compared with the total effect of 150 μg/mL of 3-*n*-butylphthalide **2** and 150 μg/mL of one azole. The sum of the inhibition rates in all cases is greater than the sum of the percent inhibition of both compounds at the concentration of 150 μg/mL and 300 μg/mL. As explained in the Introduction and illustrated in Figure 1, the mechanism of action of 3-*n*-butylphthalide is different from fluconazole.

## 3. Materials and Methods

### 3.1. General

The compounds’ purity was checked by thin layer chromatography on silica gel (DC-Alufolien Kieselgel 60 F254, Merck, Darmstadt, Germany) with methylene chloride:methanol (95:5) as an eluent. Compounds were detected by spraying the plates with 1% Ce(SO_4_)_2_, 2% H_3_[P(Mo_3_O_10_)_4_] in 10% H_2_SO_4_, followed by heating to 120 °C. Preparative column chromatography (SiO_2_, Kieselgel 60, 230–400 mesh, 40–63 μm, Merck) was performed with the application of dichloromethane:methanol (95:5) or hexane/isopropanol/acetone/ethyl acetate (60:3:1:1) as an eluent. Gas chromatography analysis (GC, FID, carrier gas H_2_) was carried out on Agilent Technologies 7890N (GC System, Santa Clara, CA, USA) with the HP-5 column (cross-linked methyl silicone, 30 m × 0.32 mm × 0.25 μm, Santa Clara, CA, USA). 1 H NMR and 13C NMR spectra were recorded in CDCl3 or DMSO solution on a Bruker Avance 600 (600 MHz, Billerica, MA, USA) spectrometer. Gas chromatography and mass spectra analysis were carried out on a Varian 2000 (Varian, Las Vegas, NV, USA).

### 3.2. Chemical Synthesis of Phthalide Lactones ***2–4***

Step 1: In a 250 mL binary flask equipped with a pressure equilibrating dropping funnel, a magnetic stirrer core, a reflux condenser fitted with a tube of anhydrous CaCl_2_, 0.0052 moles of magnesium chips, and several iodine crystals were placed. The mixture was heated, stirred continuously with a magnetic stirrer in order to transform the iodine into a gaseous state, and then cooled to room temperature. Using a dropper, 10 mL of anhydrous ethyl ether freshly distilled from LiAlH_4_ was placed in the flask. Then, 0.0062 moles of the corresponding alkyl bromide in ethyl ether was cooled for 1 h, stirring continuously with a magnetic stirrer. If necessary, in order to initiate the magnesium-organic compound formation reaction, the reaction temperature was slightly increased with the stirrer heating plate. After adding the entire contents of the dropper, the mixture was heated by stirring at the boiling point of the solvent for 2 h. After cooling the reaction mixture, 0.0025 moles of CdCl_2_ were added and the amount of anhydrous ethyl ether was increased to a total volume of 100 mL. The reaction mixture in the flask was heated until the organo-magnesium compound was completely reacted to give the organo-cadmium compound. After cooling the whole mixture again, 0.005 moles of the corresponding anhydride were added and then heated with continuous stirring for 6 h. After completion of the reaction, 50 mL of 10% hydrochloric acid was added to the reaction mixture and stirred for 2 h. The ethyl ether was evaporated using a rotary evaporator and the reaction residue was extracted in a separatory funnel with methylene chloride. The collected organic layer was dried over anhydrous MgSO_4_, then filtered through filter paper and evaporated.

Step 2a: The concentrated mixture was dissolved in 100 mL of dried THF, and then 0.006 moles of NaBH_4_ was added. The whole mixture was stirred under a reflux condenser for 8 h. The mixture was acidified with 10% HCl and stirring continued. The solvent was then evaporated on a rotary evaporator and the residue was extracted in a separatory funnel with methylene chloride. The residue was purified by column liquid chromatography.

Step 2b: The mixture was dissolved in 100 mL of benzene or toluene and then an equal (in terms of 100% yield) amount of p-toluenesulphonic acid was added and heated using a heating mantle in a round-bottomed flask fitted with an azeotropic cap, until the substrate was completely reacted, i.e., until water ceased to evaporate. The compounds obtained were purified by liquid column chromatography. Preparative column chromatography (SiO_2_, Kieselgel 60, 230–400 mesh, 40–63 μm, Merck) was performed with the application of hexane/isopropanol/acetone/ethyl acetate (60:3:1:1) as an eluent.

3-*n*-Butylphthalide (**2**) was prepared according to the procedure (steps 1, 2a), and phthalic anhydride (7.5g, 0.05 mole) was used as the substrate. 4.65 g (yield 48.8%) of phthalide was obtained. Spectroscopic data: (Appendix A):^1^H NMR (500 MHz, CDCl3), δ (ppm): 0,90 (t, 3, *J* = 7.2, CH_3_-11), 1.31–1.44 (m, 2, CH_2_-10), 1.44–1.54 (m, 2, CH_2_-9), 1.65–1.80 (m, 1, one of CH_2_-8), 1.99–2.12 (m, 1, one of CH_2_-8), 5.46 (dd, 1, *J* = 7.9, 3.7, CH-3), 7.44 (d, 1, *J* = 7.7, CH-4),7.52 (t, 1, *J* = 7.5, CH-6), 7.66 (t, 1, *J* = 7.5, CH-5),7.88 (d, 1, *J* = 7.7, CH-7). ^13^C NMR (151 MHz), δ (ppm): 13.99 (C-11), 22.55 (C-10), 27.00 (C-9), 34.57 (C-8), 81.57 (CH-3), 121.85 (CH-4), 125.83 (CH-7), 126.19 (C-7a) 129.14 (CH-6), 134.05 (CH-5), 150.27 (C–3a), 170.82 (C-1), GC-EIMS 191 (M + 1).

3-Butylidenephthalide (**3**) was prepared according to the procedure (steps 1, 2b), and phthalic anhydride (4.5 g, 0.05 mole) was used as the substrate. 3.30 g (yield 35.1%) of phthalide was obtained. Spectroscopic data: (Appendix A): ^1^H NMR (500 MHz, CDCl_3_), δ (ppm): 0.99 (t, 3, *J* = 7.4, CH_3_-11), 1.55 (m, 2, CH^2^-10), 2,45 (q, 2, *J* = 15.0, 7.5, CH^2^-9), 5.64 (t, 1, *J* = 7.8, CH-8), 7,5 (t, 1, *J* = 7.3, CH-5), 7.62–7.69 (m, 2, CH-4), 7.62–7.69 (m, 2, CH-6), 7.88 (d, 1, *J* = 7.7, CH-7). ^13^C NMR (151 MHz), δ (ppm): 13.95 (C-11), 22.86 (C-10), 27.94 (C-9), 109.60 (CH-8), 119.76 (CH-4), 124.62 (C-7a), 125.38 (C-7), 129.46 (CH-6), 134.33 (CH-5), 139.73 (C-3a), 145.90 (C-3), 167.34 (C-1) GC-EIMS 189 (M + 1).

3-Butyl-4,5,6,7-tetrahydrophthalide (**4**) was prepared according to the procedure (steps 1, 2a), 3,4,5,6-tetrahydrophthalic anhydride (7.7 g, 0.05 mole) was used as the substrate. 4.37 g (yield 45.1%) of phthalide was obtained. Spectroscopic data: (Appendix A): ^1^H NMR (500 MHz, CDCl_3_), δ (ppm): 0,88 (t, 3, *J* = 7.1, CH_3_-11), 1.22–1.,40 (m, 4, CH_2_-10, CH_2_-9), 1.41–1.50 (m, 1, one of CH_2_-8), 1.59–1.77 (m, 4, CH_2_-5, CH_2_-6), 1.79–1.87 (m, 1, one of CH_2_-8), 2.13–2.24 (m, 4, CH_2_-4, CH_2_-7), 4.74–4.79 (m, 1, CH-3). ^13^C NMR (151 MHz), δ (ppm): 13.77 (CH_3_-11), 19.77 (CH_2_-7), 21.66 (CH_2_-5), 21.69 (CH_2_-6), 22.52 (CH_2_-10), 23.25 (CH_2_-4), 26.66 (CH-9), 32.08 (CH-8), 83.01 (C-3), 126.49 (C-7a),163.84 (C-3a), 173.85 (C-1). GC-EIMS 195 (M + 1).

### 3.3. Chemical Synthesis of Fluconazole Analogues ***5–15***

#### 3.3.1. Synthesis of 1,3-(1,2,3-Triazol)propan-2-ol (**13–15**)

Synthesis of the sodium salt of 1,2,3-triazole: In a 250 mL beaker, 8.80 g of NaOH was dissolved in 30 mL of water. In a flask, 15.20 g of triazole was weighed out, then the resulting NaOH solution was added to it. The water was evaporated using an evaporator to obtain the sodium salt as a white powder.

Synthesis of 1,3-di(triazolyl)propan-2-ol: 12.9 g of 1,3-dichloro-2-propanol was added to a flask containing the entire 1,2,3-triazole sodium salt obtained, followed by 150 mL of acetone. After being provided with a reflux condenser, the flask was placed in an oil bath. The solution was stirred for 4 days at room temperature, and then the temperature was raised and heated to boiling for 24 h. The boiling solution was filtered under reduced pressure on a Schott funnel, and the precipitate was washed with 10 mL of acetone. Then, the filtrate was evaporated on an evaporator. A white precipitate was obtained, which was dried. The compounds obtained were purified by silica gel liquid column chromatography. Preparative column chromatography (SiO_2_, Kieselgel 60, 230–400 mesh, 40–63 μm, Merck) was performed with the application of dichloromethane:methanol (95:5) or hexane/isopropanol/acetone/ethyl acetate (60:3:1:1) as an eluent.

1,3-di(triazolyl)propan-2-ol **13**–**15** 6.17 g (yield 31.8%), **13** (0.71 g), **14** (3.24 g), **15** (2.22 g).

**1,3–di(1,2,3-triazol-2-yl)propan-2-ol** (**13**) (Appendix A): ^1^H NMR (500 MHz, CDCl_3_) δ = 7.79 (s, 2H), 5.66 (d, 1H, *J* = 5.5 Hz, >CH**)**, 4.47–4.55 (m, 4H, >CHCH**_2_** -N_2_ <) ppm. ^13^C NMR (500 MHz, CDCl_3_), δ = 134.67 (triazol-2-yl (C_4_, C_5_)), 69.24 (-CH(OH)-), 57.0 (-CH_2_ -) ppm. GC-EIMS 195 (M + 1).

**1-(1,2,3 triazol-1-yl)-3-(1,2,3-triazol-2-yl)propan-2-ol** (**14**) (Appendix A): ^1^HNMR (500 MHz, DMSO-d6), δ = 8.08 (s, NC_4_ H), 7.8 (s, 2H, NC_4_ H, NC_5_ H**)** 7.72 (s, 1H, NC_5_ H), 5.55 (d, 1H, *J* = 5.5 Hz, >CH**)**, 4.53 (dd, 2H *J* = 13.25 Hz, *J* = 5.5 Hz, -CH**_2_**-), 4.34 (dd, 2H *J* = 13.25 Hz, *J* = 5.5 Hz, -CH**_2_**-) ppm. ^13^C NMR (500 MHz, DMSO-d6), δ = 134.94 ppm (1,2,3-triazol-2-yl (C_5_, C_4_)), 133.50 (1,2,3-triazol-1-yl (C_4_)), 126.28 (1,2,3-triazol-1-yl (C_5_)), 68.93 (-CH(OH)-), 57.96 (-CH_2_-N_2_ <), 53.29 (-CH_2_-N_1_ <) ppm. GC-EIMS 195 (M + 1).

**1,3–di(1,2,3-triazol-1-ylo)propan-2-ol (15)** (Appendix A)**:** ^1^H NMR (500 MHz, DMSO-d6), δ =8.08 (s, NC_4_ H), 7.72 (s, 1H, NC_5_ H), 5.64 (d, 1H, *J* = 5.5 Hz, >CH**)**, 4.53 (dd, 2H *J* = 13.25 Hz, *J* = 5.5 Hz, -CH**_2_**-), 4.34 (dd, 2H *J* = 13.25 Hz, *J* = 5.5 Hz, -CH**_2_**-) ppm. ^13^C NMR (500 MHz, DMSO-d6), δ =133.57 (1,2,3-triazol-1-yl (C_4_)), 126.26 (1,2,3-triazol-1-yl (**C_5_**)), 68.83 (-CH(OH)-), 53.26 (-CH_2_-) ppm. GC-EIMS 195 (M + 1).

**1,3-bis(1H-1,2,3-triazol-1-yl)-2-phenylpropan-2-ol (5):** Synthesis of 1,3-dichloro-2-(phenyl)propan-2-ol was performed according to a known method [43]: 1,3-dichloroacetone (25.40 g, 0.200 mol) and dry diethyl ether (100 mL) was placed in two-necked flask equipped with stirring bar. The flask, under slow flow of nitrogen gas, was capped with rubber septa and then was placed in cooling bath (−60 °C). The freshly prepared phenylmagnesium bromide (obtained in the reaction of 5.35 g magnesium (0.22 mol) with 31.4 g (0.20 mol) phenyl bromide in 80 mL of dry diethyl ether) was added into the solution via a syringe for 90 min. After completion of Grignard reagent addition, the reaction mixture was stirred for the next 1 h. Then, a solution of glacial acetic acid (13.0 g, 0.22 mol) in 20 mL of ethyl ether was added in a few portions. The reaction mixture was left to reach 0 °C (1 h) and then water (100 mL) was added slowly (15 min), keeping the temperature in the range 0–5 °C. After the disappearance of the solid phase, the reaction mixture was stirred for an additional 15 min. The organic layer was separated and residual water solution was washed with ethyl ether (3 × 100 mL). Combined etheric phases were dried over anhydrous magnesium sulphate. Solvents were removed using a rotary evaporator and 5.75 g of oily residue was used for alkylation of 1,2,3-triazole without further purification.

Sodium 1,2,3-triazolate (prepared according to the previously described procedure using 5.52 g of 1,2,3-triazole and 3.20 g NaOH) was suspended in 30 mL of acetonitrile in a round bottom flask and then a solution of crude 1,3-dichloro-2-(phenyl)propan-2-ol (5.75 g) in acetonitrile (20 mL) was added. The mixture was stirred at room temperature for 3 days and then the flask was equipped with a condenser and the mixture was refluxed for the next 3 days. The hot reaction mixture was filtered, and solid residue was washed three times with hot acetonitrile (10 mL). The filtrate was evaporated to dryness using a rotary evaporator. Resulting oil was extracted with CCl_4_ (5 × 100 mL). Solid residue was treated with 15 mL of acetonitrile and undissolved matter was filtered off. Acetonitrile was removed under reduced pressure and the obtained solid was purified by crystallization from water. The colorless crystalline product was isolated with 4.2 g (yield 55.6%) of three regioisomers. Then, preparative column chromatography (SiO_2_, Kieselgel 60, 230–400 mesh, 40–63 μm, Merck) was performed with the application hexane/isopropanol/acetone/ethyl acetate (60:3:1:1) as an eluent; 0.47 g of 1,3-bis(1H-1,2,3-triazol-1-yl)-2-phenylpropan-2-ol (**5)** was obtained. Spectroscopic data (Appendix A): ^1^H NMR (500 MHz, DMSO-d6), δ =7.75 (s, NC_4_ H), 7. 59 (s, 1H, NC_5_ H), 7.37 (m, 2H- phenyl) 7.21 (m, 4H-phenyl) 6.17 (m, 1H, Hz, >CH**)**, 4.93 (dd, 2H *J* = 13.25 Hz, *J* = 5.5 Hz, -CH**_2_-**), 4.35 (dd, 2H *J* = 13.25 Hz, *J* = 5.5 Hz, -CH**_2_**-) ppm. ^13^C NMR (500 MHz, DMSO-d_6_), δ =140.71 (C-1-phenyl) 133.19 (1,2,3-triazol-1-yl (C_4_)), 128.28 and 127.05 (C_2–5_-phenyl) 126.26 (1,2,3-triazol-1-yl (C_5_)), 75.25 (-CH(OH)-), 57.25 (-CH_2_-) ppm. GC-EIMS 271 (M + 1).

#### 3.3.2. General Methods of Synthesis of Ester: **6**–**10**

To one isolated 1,3-di(1,2,3-triazole-2-yl)propan-2-ol (**13**–**15**) dissolved in 25–50 mL of dry dichloromethane and 3 equivalents of pyridine and 1.2 equivalents, one of phenylacetyl chloride, 2-thiophenecarbonyl chloride or adipyl chloride was added. The solution was stirred on a magnetic stirrer to react to all the substrate. After the synthesis, the solution was washed with 5 mL of 5% HCl and 1% NaCl. The water layer was extracted 3 times by dichloromethane. The extracted solution was dried with MgSO_4_ and concentrated in an evaporator. The product obtained was a dark yellow, oily liquid. Preparative column chromatography (SiO_2_, Kieselgel 60, 230–400 mesh, 40–63 μm, Merck) was performed with the application of dichloromethane: methanol (95:5) or hexane/isopropanol/acetone/ethyl acetate (60:3:1:1) as an eluent.

**1,3-di(1,2,3-triazol-2-yl)propan-2-yl phenylacetate (6)** was prepared according to Section 3.3.2. 1,3-di(1,2,3-triazol-2-ylo)propan-2-ol (13) (0.2 g, 0.001 mole) and phenylacetyl chloride (0.18 g, 0.0012 moles) was used as the substrate. 0.24 g (yield 76%) of ester (**6**) was obtained. Spectroscopic data (Appendix A): ^1^H NMR (500 MHz, CDCl_3_) δ = 7.57 (s, 2H, triazol-2-yl (C_4_, C_5_)), 7.22–7.25 (m, 3H, ph), 7.10–7.16 (m, 2H, ph), 5.66–5.80 (m, 1H, >CH), 4.79 (dd, 2H *J* = 13.25 Hz, *J* = 5.5 Hz, -CH**_2_**-), 4.63 (dd, 2H *J* = 13.25 Hz, *J* = 5.5 Hz, -CH**_2_**-N_2_ <), 3.49 (s, 2 H, -O-CH_2_ ph) ppm. ^13^C NMR (500 MHz, CDCl_3_), δ = 170.17 (C=O), 134.67 (triazol-2-yl (C**_4_,** C_5_)), 133.19 (C_1_ ph), 129.31 (C, C_26_ ph), 128.47 (C_2_, C_6_ ph), 127.104 (C_4_ ph), 70.29 (-CH(OH)-), 54.60 (-CH_2_ -), 48.84 (-CH_2_ ph) ppm. GC-EIMS 313 (M + 1).

**1-(1,2,3-triazol-1-yl)-3-(1,2,3-triazol-2-yl)propan-2-yl phenylacetate (7)** was prepared according to Section 3.3.2. 1-(1,2,3 triazol-1-yl)-3-(1,2,3-triazol-2-yl)propan-2-ol (**14**) (0.25 g, 0.0013 mole) and phenylacetyl chloride (0.23 g, 0.0015 mole) were used as the substrate. 0.28 g (yield 70%) of ester (**7**) was obtained. Spectroscopic data: Appendix A: ^1^HNMR (500 MHz, DMSO-d6), δ = 7.61 (s, NC_4_ H), 7.58 (s, 2H, NC_4_ H, NC_5_ H) 7.29 (s, 1H, NC_5_ H), 7.22–7.25 (m, 3H, ph), 7.10–7.16 (m, 2H, ph), 5.61–5.71 (m, 1H) >CH, 4.74 (dd, 2H *J* = 13.25 Hz, *J* = 5.5 Hz, -N_1_ CH_2_-), 4.56 (dd, 2H *J* = 13.25 Hz, *J* = 5.5 Hz, -CH**_2_**-N_2_ <), 3.54 (s, 2H, -O-CH_2_ ph) ppm. ^13^C NMR (500 MHz, CDCl_3_), δ = 170.16 (C=O), 134.67 (1,2,3-triazol-2-yl (C_4_, C_5_)), 133.74 (1,2,3-triazol-1-yl (C_4_)),133.19 (C_1_ ph), 129.31 (C_2_, C_6_ ph), 128.47 (C_2_, C_6_ ph) 127.10 (C_4_ ph), 124.70 (tetrazol-1-yl (C_5_)), 70.19 (-CH(OH)-), 54.13 (-N_2_ CH_2_-), 50.06 (-N_1_ CH_2_-), 41.13 (-CH_2_ ph) ppm. GC-EIMS 313 (M + 1).

**1-(1,2,3-triazol-1-yl)-3-(1,2,3-triazol-2-yl)propan-2-yl 2-tiophenylacetate** (**8**) was prepared according to Section 3.3.2. 1-(1,2,3 triazol-1-yl)-3-(1,2,3-triazol-2-yl)propan-2-ol (**14**) (0.22 g, 0.0013 mole) and 2-tiophenecarbonyl chloride (0.32 g, 0.0016 mole) were used as the substrate. 0.28 g (yield 70.9%) of ester (**8**) was obtained. Spectroscopic data (Appendix A): ^1^HNMR (500 MHz, CDCl_3_), δ = 7.73 (s, NC_4_ H), 7.72 (s, 2H, NC_4_ H, NC_5_ H) 7.71 (s, 1H, NC_5_ H), 7.56–7.62 (m, 2H, thiophenyl), 7.06–7.08 (m, 1H, thiophenyl), 5.61–5.80 (m, 1H, >CH), 4.74 (two dd, 2H *J* = 12.05 Hz, *J* = 6.0 Hz, -N_1_ CH_2_-), 4.56 (two dd, 2H *J* = 15.15 Hz, *J* = 6.0 Hz, -CH**_2_** -N_2_ <), ppm. ^13^C NMR (500 MHz, CDCl_3_), δ = 160.55 (C=O), 135.08 (-S-C-C_5_-) 135.07 (N_2_ C_4_, C_5_), 134.64 (-S-C_1_-), 133.10 (**C**_3_-S-C_1_-), 131.85 (**C**_4_-C_3_-S-), 128.47 (N C_1**4**_), 124.86 (N_1_
**C_5_**), 70.63 (-CH(OH)-), 54.17 (-N_2_ CH_2_-), 50.25 (-N_1_ CH_2_-) ppm. GC-EIMS 305 (M + 1).

1-(tetrazol-1-yl)-3-(tetrazol-2-yl)propan-2-yl 2-tiophenylcarboxylate (**9**) was prepared according to Section 3.3.2. Crude mixtures of three regioisomers—1,3–di(tetrazol-2-ylo)propan-2-ol 1-(tetrazol-1-yl)-3-(tetrazol-2-yl)propan-2-ol and 1,3–di(tetrazol-1-ylo)propan-2-ol) prepared according to a known method [44,45]) (0.34 g) and 2-tiophenecarbonyl chloride (0.32 g, 0.0016 mole)—were used as the substrate. 0.15 g (yield 32.6%) of ester (9) was obtained. Spectroscopic data (Appendix A): ^1^HNMR (500 MHz, DMSO-d6), δ = 9.46 (s, 1H, N_2_ C_5_ H) 9.01 (s, 1H, N_1_ C_5_ H), 7.95 (s, 1H, -S-C-C_5_ H-), 7.61 (s, 1H, -C_3_ H-S-C_1_), 7.15 (s, 1H, -S-C_4_ H-), 5.78–5.83 (m, 1H, >CH), 5.35 (two dd, 2H *J* = 13.01 Hz, *J* = 8.5 Hz, -N_1_ CH_2_ -), 5.13 (two dd, 2H *J* = 12.15 Hz, *J* = 8.5 Hz, -CH_2_ -N_2_ <), 3.54 (s, 2 H, -O-CH_2_ ph) ppm. ^13^C NMR (5), 131.45 (C_3_-S-C_1_-), 128.79 (C_4_-C_3_-S-), 70.45 (-CH(OH)-), 53.07 (-N_2_ CH_2_-), 48.55 (-N_1_ CH_2_-) ppm. GC-EIMS 307 (M + 1).

**1,6-bis[1-(1,2,3-triazol-1-yl)-3-(1,2,3-triazol-1-yl)propan-2-yl] adipinate (10)** was prepared according to Section 3.3.2. 1,3-di(1,2,3-triazol-1-ylo)propan-2-ol (**15**) (0.2 g, 0.001 mole) and adipyl chloride (0.110 g, 0.0006 mole) were used as the substrate. 0.25 g (yield 50.2%) of ester (**10**) was obtained. Spectroscopic data (Appendix A): ^1^H NMR:: δ 1.12 (4H, quint, *J* = 3.2 Hz), 2.05 (4H, t, *J* = 6.7 Hz), 4.56–4.62 (4H, 4.58 (d, *J* = 8.2 Hz), 4.61 (d, *J* = 8.2 Hz)) 4.77–4.83 (4H, 4.79 (d, *J* = 3.8 Hz), 4.8 (d, *J* = 3.8 Hz), 5.56 (2H, quint, *J* = 3.7 Hz), 7.72 (4H, d, *J* = 0.7 Hz), 8.12 (4H, d, *J* = 0.7 Hz).^13^C NMR (500 MHz, DMSO-d6),: δ 23.2(2C), 32.7 (2C), 49.8 (4C), 70.0 (2C), 125.8 (4C), 133.5 (4C), 171.3(1C).

**1-[2-(2H-tetrazol-2-ylmethyl)prop-2-en-1-yl]-1H-tetrazole (11):** In a 50 mL flask, 25 mL of methanol was placed, and 1.46 g (0.018 mol) of sodium was added. After the Na had reacted completely, 4.50 g (0.064 mol) of tetrazole was introduced. Stripping the solvents in a vacuum evaporator gave the sodium salt of tetrazole. 30 mL of acetonitrile and 3.72 g (0.03 mol) of 3-chloro-2-(chloromethyl) prop-1-ene were added to the obtained salt. The contents of the flask were heated with constant stirring to gentle reflux for 4 days.

After the completion of the reaction, the mixture was filtered off the sodium chloride formed and concentrated in a vacuum evaporator. The obtained slightly yellow oil (6.2 g) was separated by chromatography. Liquid chromatography was performed on a silica gel column (70–230 mesh). With the eluent CH_2_Cl_2_/MeOH 10/0.25–0.6 (*v/v*) 2.92 g was obtained (yield of 11 50.7%). Preparative column chromatography (SiO_2_, Kieselgel 60, 230–400 mesh, 40–63 μm, Merck) was performed with the application of dichloromethane: methanol (95:5) or hexane/isopropanol/acetone/ethyl acetate (60:3:1:1) as an eluent. Spectroscopic data:(Appendix A): ^1^HNMR CDCl_3_, (δ = 8.81 (s, 1H), 8.60 (s, 1H), 5.34 (s, 1H), 5.28 (s, 2H), 5.25 (s, 1H), 5.05 (s, 2H) ppm; ^13^CNMR CD_3_CN, δ = 154.25, 144.77, 137.08, 122.10, 55.82, 50.82 ppm. GC-EIMS 193 (M + 1).

**1,3-di(tetrazol-2-yl)-2-chloromethylpropane (12):** In a 100 mL flask, 30 mL of methanol was placed, and 1.15 g (0.05 mol) of sodium was added. After the sodium had reacted completely, 3.50 g (0.05 mol) of tetrazole was introduced. Stripping the solvent in a vacuum evaporator gave the sodium salt tetrazole. 60 mL of acetonitrile and 4.0 g (0.02 mol) of 2-[3-chloro-2-(chloromethyl) propyl]-2H-tetrazole were introduced into the obtained salt. The contents of the flask were stirred and heated to gentle reflux for 10 days under reflux.

After the completion of the reaction, the mixture was filtered off the sodium chloride formed and concentrated in a vacuum evaporator. The obtained slightly yellow oil, 5.1 g with 0.85 g (yield 18.8%). 1,3-di(tetrazol-2-yl)-2-chloromethylpropane (**12**), was separated by chromatography. Liquid chromatography was performed on a silica gel column (70–230 mesh). Eluent: CH_2_Cl_2_/MeCN/MeOH 10/1.0/0.15–1.2 (*v/v*). Spectroscopic data:(Appendix A) ^1^HNMR CD_3_CN, δ = 8.62 (s, 3H), 4.85 (d, 6H), 3.71 (m, 1H) ppm; ^13^CNMR CD_3_CN, δ = 154.87, 53.17, 40.18 ppm. GC-EIMS 227 (M + 1).

### 3.4. In Silico Methods

The prediction of toxicity was based on calculations from an algorithm developed by the Swiss Institute of Bioinformatics at the University of Lausanne. A computer-aided drug design (CADD) algorithm based on machine learning can predict the biological properties of a test chemical with high probability [41]. The detailed methodology of the program is described by the authors of Swiss ADME.

Oral toxicity in the rat was developed using the way2drug model created by the Department for Bioinformatics, Laboratory for Structure-Function Based Drug Design, Institute of Biomedical Chemistry (IBMC) in Moscow [42].

The structures of all the chemical compounds used in this work, including fluconazole and phthalide lactones, were input into the above programs. Swiss ADME allowed us to obtain information concerning the lipophilicity of the above compounds and information concerning the probable inhibition of key cytochrome P450 enzymes responsible for metabolism in the human body. A determination of 50% of the lethal dose for the rat after oral administration of the compounds under study was obtained using the way2drug program, which calculated theoretical toxicity in a similar way to the previous program.

### 3.5. Method for Determination of Biological Activity

Growth conditions of microorganisms: The frozen stocks were cultured in 300 mL Erlenmeyer flasks. Each strain was inoculated under sterile conditions in Sabouraud medium (75 mL) at 35 °C for 24 h on a rotary shaker.

Optical density measurements: Measurement of the optical density of the cultures on the plate, as well as the maintenance of optimum temperature throughout the process of culturing the microorganisms, was possible thanks to the use of a BioTek TS-800 microplate reader (Riga, Latvia) and the Gen5 computer software as its controller (Figure 16).

The Gen5 program automated the process of turbidimetric measurement of the optical density of each sample, collecting and analyzing the obtained turbidity values. After culture, the software generated growth curves for each culture (well) on the plate taking into account the OD values of the reference samples.

#### 3.5.1. Antifungal Activity

The antifungal activity of all the obtained compounds was determined by determining the minimum inhibitory concentration (MIC) according to the formula:$$\mathrm{MIC}=\frac{\mathrm{OD}\;\mathrm{control}-\mathrm{OD}\;\mathrm{sample}\;}{\mathrm{OD}\;\mathrm{control}}\times 100\%$$
by the broth microdilution method according to CLSI guidelines [54] with modifications. Sabouraud medium was used for the study. Active compounds were prepared in dimethylsulfoxide solution and then diluted in Sabouraud broth to obtain final microtiter plate concentrations in the range of 50–350 μg/mL. The fluconazole solution was tested in the range of 0.195–100 μg/mL.

Then, 50 μL of each solution was pipetted into a 96-well microtiter plate. The inoculum was standardized to a 0.5 McFarland grade standard in 0.9% NaCl solution and then diluted to give a final suspension with a cell density of 0.5 × 10^3^ to 2.5 × 10^3^ CFU/mL. The inoculum amount used was 50 μL. The reference sample was DMSO diluted in broth without the addition of inoculum and the negative control was inoculated analogue solution. All samples were tested in triplicate. Microtiter plates were incubated at 37 °C for 18 h until optical density (OD) = 1.0 in control samples on a Biosan PST-60 HL ThermoShaker (Riga, Latvia) shaker at 1200 rpm. The fungistatic activity of the compounds was evaluated by measuring the absorbance at 595 nm on a BioTek 800 TS to determine the MIC_50_ and MIC_90_ values (i.e., the concentration of compound required to inhibit the growth of the microorganism by 50% and 90% compared to the control sample) (Figure 16).

#### 3.5.2. Fractional Inhibitory Concentration

To evaluate the combined effect of the compounds, a checkerboard array method was performed based on the method described in “*Antibiotics in Laboratory Medicine*” [52]. Sabouraud medium was used for the study. Representative compounds exhibiting low MIC values were prepared in DMSO solution and then diluted directly on a microtiter plate with broth to give final concentrations in the range of 0.39–200 μg/mL and 3-*n*-butylphthalide in the range of 1.563–100 μg/mL in the mixture. The reference sample was DMSO diluted with Sabouraud medium without the addition of inoculum. Controls were samples containing only one test compound in the range of 1.563–100 μg/mL and 3-*n*-butylphthalide in the range of 3.125–200 μg/mL. The compounds were tested in the range between their MIC and progressive two-fold dilutions. All samples were tested in triplicate. Microtiter plates were incubated and then absorbance was read analogously to determine antifungal activity. A potential synergistic effect was noted when the inhibition of both compounds in the culture mixture were greater than that of the compounds alone. The fractional inhibitory concentration (FIC) was calculated using the formula [53]:$$\Sigma \mathrm{FIC}=\frac{\mathrm{MIC}\;\mathrm{of}\;\mathrm{agent}\;A\;\mathrm{in}\;\mathrm{combination}}{\mathrm{MIC}\;\mathrm{of}\;\mathrm{agent}\;A\;\mathrm{alone}}+\frac{\mathrm{MIC}\;\mathrm{of}\;\mathrm{agent}\;B\;\mathrm{in}\;\mathrm{combination}}{\mathrm{MIC}\;\mathrm{of}\;\mathrm{agent}\;B\;\mathrm{alone}}$$

## 4. Conclusions


Phthalide lactones in combination with azole compounds were effective in reducing the concentration of fluconazole and its derivatives, due to the synergistic action in the reaction system against fluconazole-resistant strains of *Candida albicans* isolated from patients and strains that did not develop resistance to fluconazole.The synergistic effects of fluconazole (**1**) and three synthetic lactones identical to naturally occurring celery plants—3-*n*-butylphthalide (**2**), 3-*n*-butylidenephthalide (**3**), 3-*n*-butyl-4,5, 6,7—tetrahydrophthalide (**4**)—against *Candida albicans* ATCC 10231, *C. albicans* ATCC 2091, *C. guilliermondii* KKP 3390 and two strains of *C. albicans* resistant to fluconazole, were compared with the activity of individual compounds separately. In all cases, these compounds showed high fungistatic activity, which makes them potential agents for use in pharmacology.Eleven azole derivatives not previously described in the literature were designed and synthesized. The structures of the compounds obtained were determined by 1HNMR and 13C NMR spectroscopy, and molecular weights were determined by GC-MS or elemental analysis.


The relationship between 3-*n*-butylphthalide and fluconazole analogues 5–10, 12 and 13 was investigated. The results of the experiment confirmed that in almost all cases, the mixture consisting of the azole derivative and 3-*n*-butylphthalide (**2**) achieved even six times lower concentration values with a simultaneous higher inhibition factor. The interaction of 3-*n*-butylphthalide with the azoles showed a synergistic effect in all cases.4.High specificity for individual *Candida* strains was also observed in all tests.5.A correlation between log P and fungistatic and synergistic effects has not been established. The drug transporters CDR1 and CDR2 play a key role in the synergistic effect.

## Figures and Tables

**Figure 1 antibiotics-11-01500-f001:**
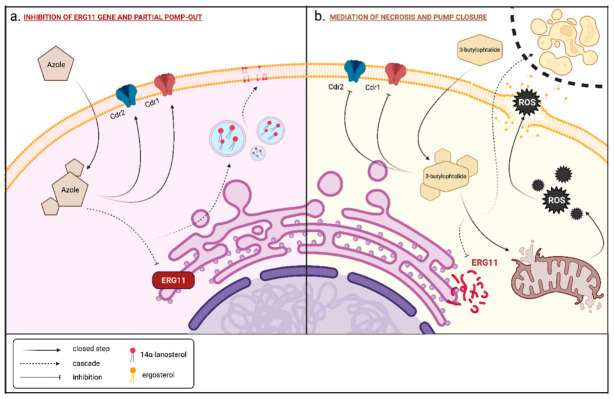
(**a**) Mechanism of action of 3-butylphalide on yeast cell metabolism. (**b**) Mechanism of effect of azole compounds on yeast cell (created in BioRender).

**Figure 2 antibiotics-11-01500-f002:**
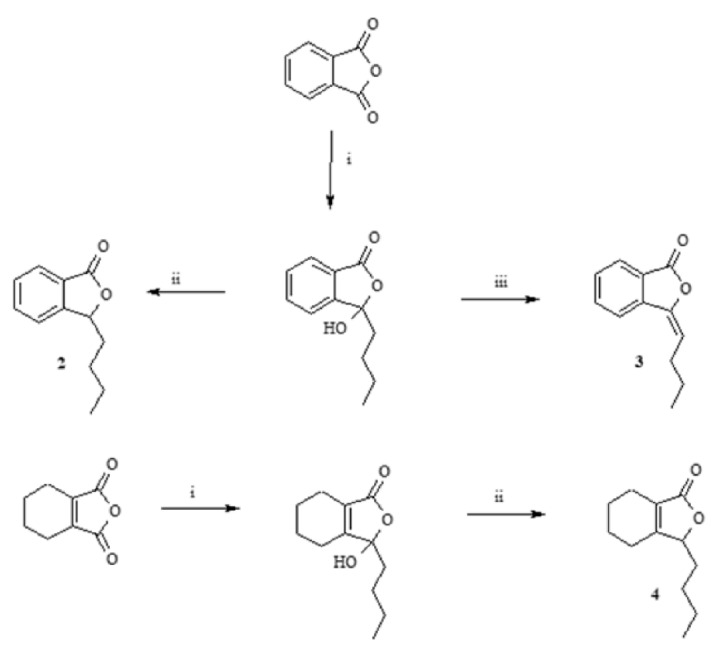
Scheme of synthesis of phthalide lactones **2–4**. i—Et_2_O, Mg, n-C_4_H_9_Br, CdCl_2_; ii—THF, NaBH_4_; iii—toluene, TsOH.

**Figure 3 antibiotics-11-01500-f003:**
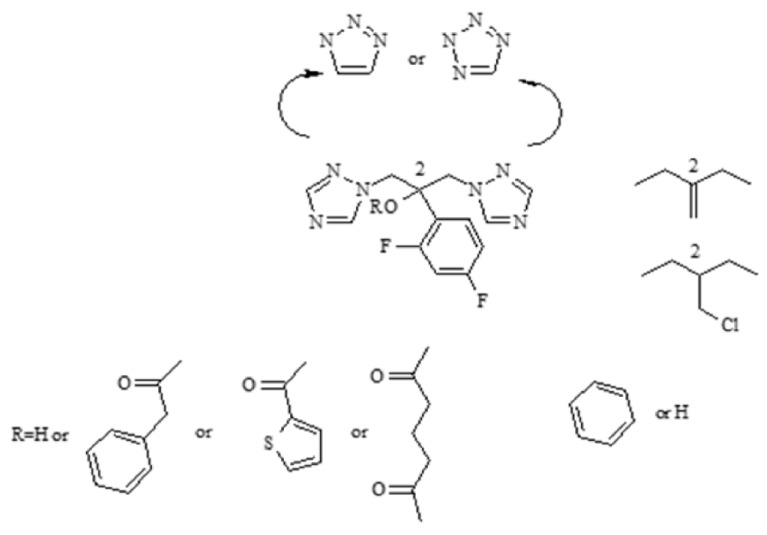
Strategy of design of fluconazole derivatives.

**Figure 4 antibiotics-11-01500-f004:**
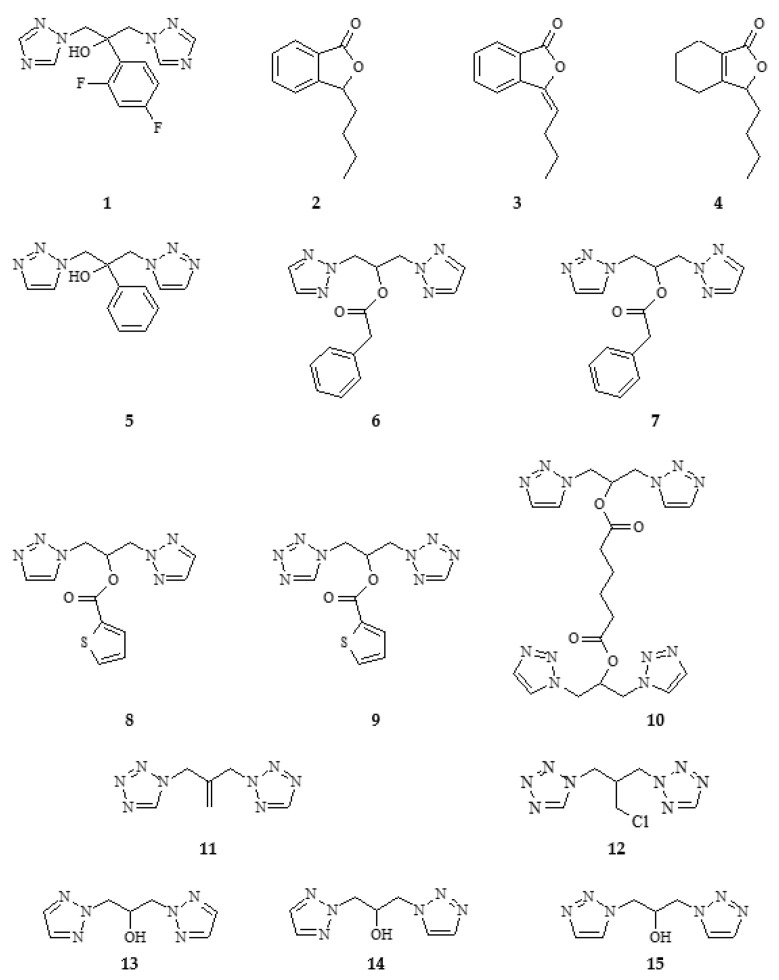
List of fluconazole (**1**), phthalide lactones **2**–**4**, and designed fluconazole analogues **5–15** used in biological tests.

**Figure 5 antibiotics-11-01500-f005:**
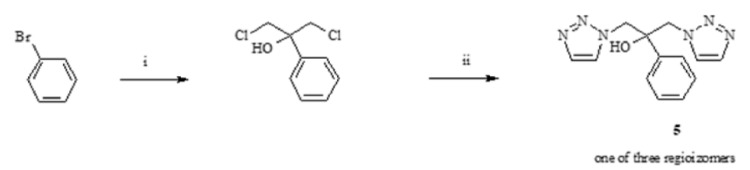
Scheme of synthesis of 1,3-di(1H-1,2,3-triazol-1-yl)-2-(phenyl) propan-2-ol (**5**). i—Et_2_O, Mg, 1,3-dichloroacetone; ii—acetonitrile, sodium 1,2,3-trazole, reflux.

**Figure 6 antibiotics-11-01500-f006:**
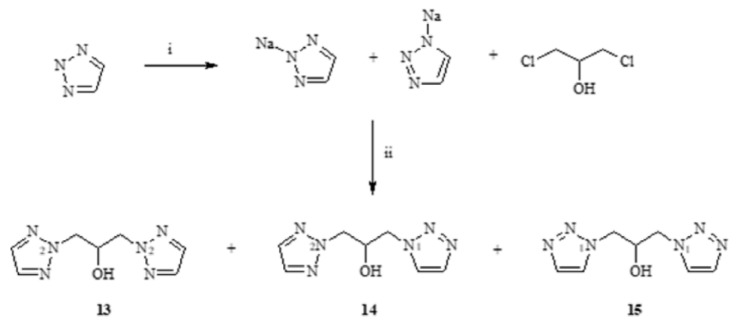
Scheme of synthesis of alcohols **13–15**. i—Na, MeOH 1,3- dichloroacetone; ii—acetonitrile, reflux.

**Figure 7 antibiotics-11-01500-f007:**
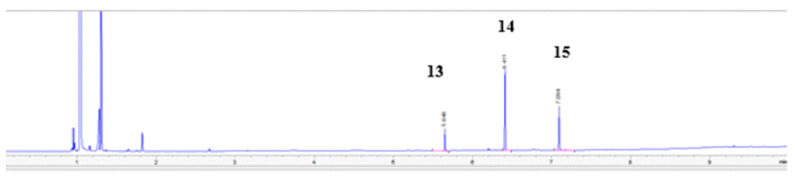
Chromatogram crude extract of mixture alcohols **13–15**.

**Figure 8 antibiotics-11-01500-f008:**
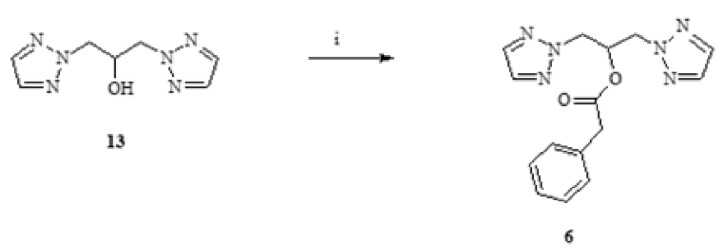
An example of ester synthesis. Scheme of synthesis of 1,3-di(1,2,3-triazol-2-yl) propan-2-yl phenylacetate (**6**). i—pyridinium, chloride of phenylacetic acid.

**Figure 9 antibiotics-11-01500-f009:**
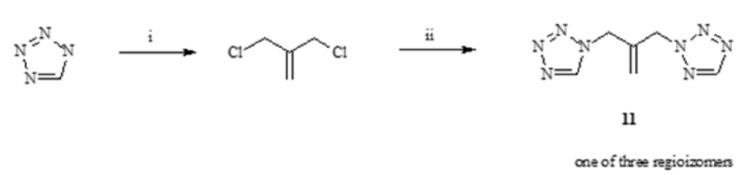
Scheme of synthesis of 1-[2-(2H-tetrazol-2-ylmethyl) prop-2-en-1-yl]-1H-tetrazole(**11**). i—sodium, methanol, ii—acetonitrile, reflux.

**Figure 10 antibiotics-11-01500-f010:**
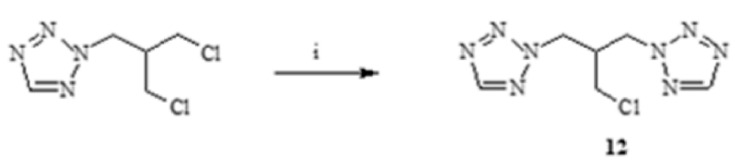
Scheme of synthesis of 1,3-di(tetrazol-2-yl)-2-chloromethylpropane (**12**). i—sodium, methanol, 1,2,3-triazole, reflux.

**Figure 11 antibiotics-11-01500-f011:**
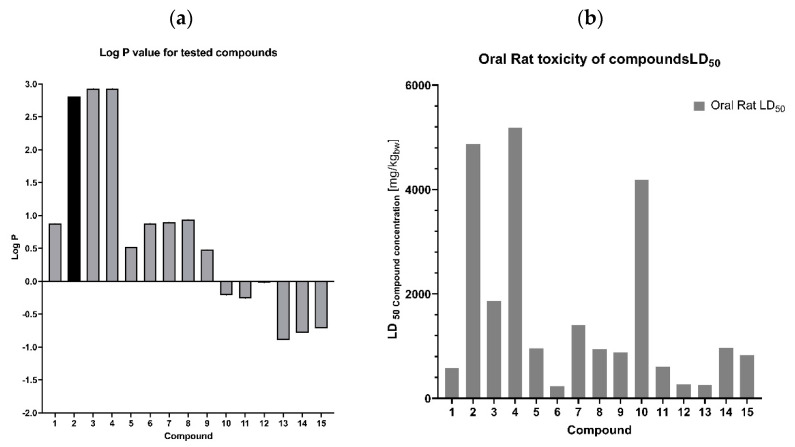
In silico prediction of lipophilicity (**a**) and toxicity (**b**) for fluconazole (**1**), lactones (**2–4**) and fluconazole analogues (**5–15**).

**Figure 12 antibiotics-11-01500-f012:**
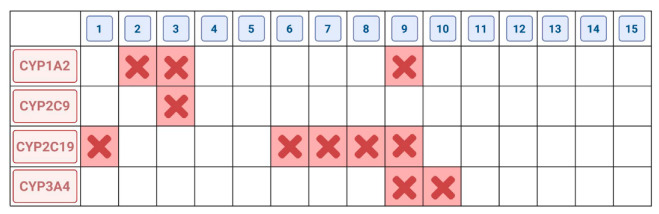
ADME analysis. CYP inhibitor prediction of fluconazole **1**, lactones **2–4**, analogues of fluconazoles **5–15** based on their chemical structure. Only **8** of them show inhibition of CYPs.

**Figure 13 antibiotics-11-01500-f013:**
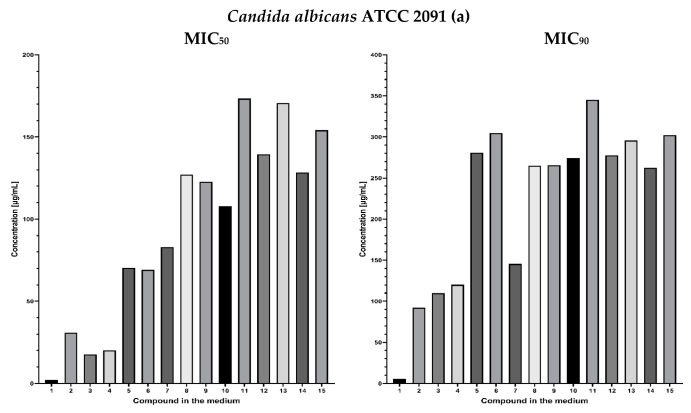
Experimentally determined fungistatic activity of fluconazole and lactones **2–4** and fluconazole analogues **5–15** derivative against *Candida albicans* ATCC 10231, *C. albicans* ATCC 2091, *C. zeylanoides* KKP 3528, and *C. guilliermondii* KKP 3390 (**a**–**e**). MIC values are in μg/mL. MIC_50_: the amount of fungistatic substance (in µg/mL) inhibiting yeast growth by 50%. MIC_90_: the amount of fungistatic substance (in µg/mL) inhibiting yeast growth by 90%.

**Figure 14 antibiotics-11-01500-f014:**
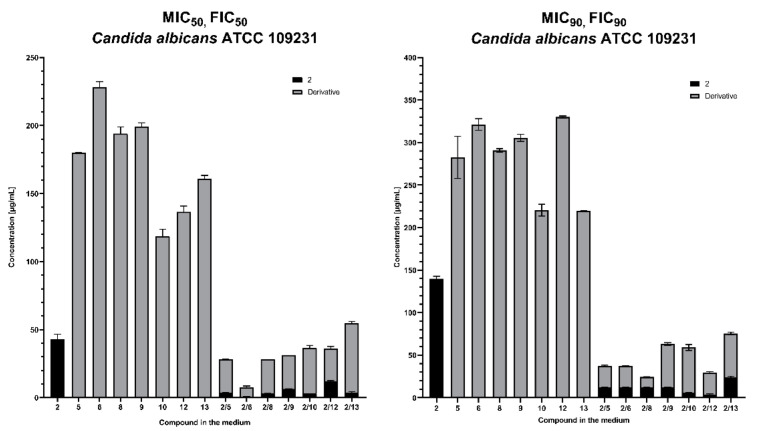
Comparison of MIC and FIC factors for selected azole derivatives and 3-*n*-butylphthalide against *C. albicans* ATCC 109231, *C. albicans* ATCC 2091, and *C. guilliermondii* KKP 3390.

**Figure 15 antibiotics-11-01500-f015:**
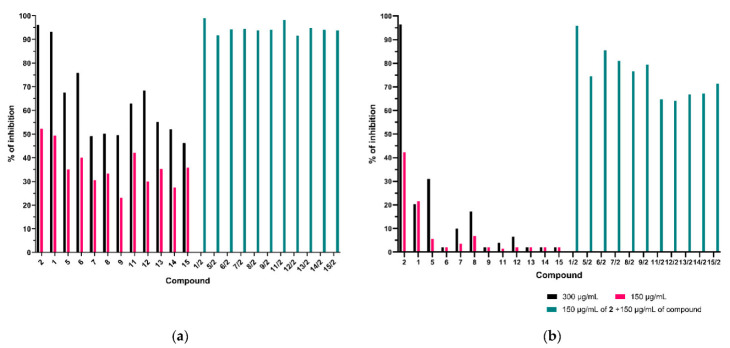
(**a**) Percent of inhibiting the growth of *Candida albicans* AM38/22 (MIC_90_ > 256 µg/mL) by lactone **2** and azoles **1–9**, **11–15** alone and in combination lactone **2** with one of the azoles. (**b**) Percent of inhibiting the growth of *Candida albicans* AM38/680 (MIC_90_ > 1024 µg/mL) by lactone **2** and azoles **1–9**, **11–15** alone and in combination lactone 2 with one of the azoles.

**Figure 16 antibiotics-11-01500-f016:**
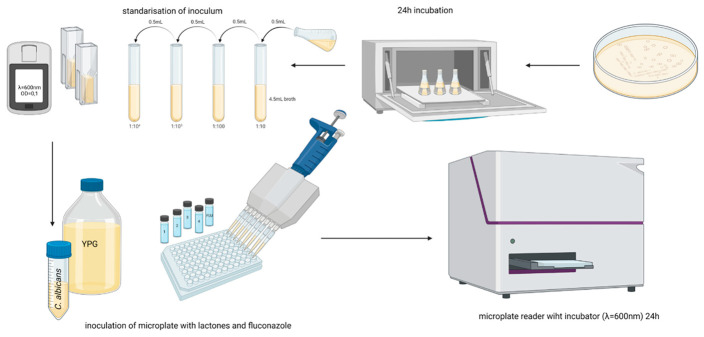
Schematic representation of the method for determination of biological activity. The standardized yeast inoculum was placed in a 96-well microplate containing the phthalide lactones at the quantified concentrations. After 24 h of incubation, the plates were placed in a microplate reader for optical density (OD) determination (Appendix A) (created in BioRender).

**Table 1 antibiotics-11-01500-t001:** Determination of the MIC in the combination and the FIC values* for the 2,3,4 phthalide lactones and fluconazole in *Candida albicans* ATCC 10231, *C. albicans* ATCC 2091, *C.*
*guilliermondii*.

Compound	ATCC 10231				ATCC 2091				KKP 3390			
	MIC_synergy_		FIC		MIC_synergy_		FIC		MIC_synergy_		FIC	
	50%	90%	50%	90%	50%	90%	50%	90%	50%	90%	50%	90%
**1**	0.78	1.56	1.41	0.27	0.78	1.56	1.17	0.57	1.56	0.78	0.31	0.28
**2**	0.78	3.13			25	25			3.13	1.26		
**1**	0.62	1.59	0.42	0.28	0.55	1.41	0.42	0.47	1.27	0.58	0.32	0.35
**3**	1.20	4.47			3.72	22.25			4.32	20.18		
**1**	0.42	1.43	0.41	0.34	0.70	1.48	0.55	0.49	1.22	6.73	0.43	0.65
**4**	1.32	15.27			4.45	20.27			11.58	24.86		

KKP 3390 culture system. MIC values are in μg/mL.

## Data Availability

The data presented in this study are available on request from the corresponding author.

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
