# Peer review of "Synergic Effect of Phthalide Lactones and Fluconazole and Its New Analogues as a Factor Limiting the Use of Azole Drugs against Candidiasis"

_antibiotics, 2022, doi:10.3390/antibiotics11111500_

Round 1

Reviewer 1 Report

Antibiotics

Article: "Synergic effect of phthalide lactones and fluconazole and its new analogues as a factor limiting use of azole drugs against candidiasis"

-        Line 111: "L. wallichii". Write in italic form.

-        Line 177: what is the chemical class of fluconazole.

-        In table 1. Write the names of the tested compounds? Give the meaning of all abbreviations in the table foot.

-        Line 379: "The synergistic effect occurring between 3-butylidenophthalide and fluconazole was interesting". What is the reason for the synergistic effect related to the chemical structures, 

- Figure 14 is not clear to read

Author Response

Dear Reviewer

I submit after revision manuscript titled ‘Synergic effect of phthalide lactones and fluconazole and its new analogues as a factor limiting the use of azole drugs against candidiasis’ for publication in the Antibiotics (Special Issue Design and Synthesis of Novel Antimicrobial Agents. The paper was coauthored by Piotr Krężel, Teresa Olejniczak, Aleksandra Tołoczko, Joanna Gach, Marek Weselski and Robert Bronisz.

We are grateful for the really meritorical evaluation. The referees comments were very helpful in the improvement of our manuscript and practically all comments were accepted. Our responses to reviewers’ comments have been included below. The changes in the revised version of the manuscript were highlighted in a different colour.

I sent you a pdf document.

Yours sincerely,

Teresa Olejniczak

Faculty of Biotechnology and Food Science

Wrocław University of Environmental and Life Sciences

Norwida 25, 50-375 Wroclaw, Poland

Reviewer 2 Report

The current manuscript is set to determine the synergic effect of phthalide lactones and fluconazole and its new analogues as a factor limiting the use of azole drugs against candidiasis. This original study provides evidence of the new analogous of lactones and fluconazole that may be more efficient in the treatment of candidiasis. Therefore, this study is important for the future treatment of candida infections. However, some remarks given below are needed to be considered and clarified:

·       It is unclear what kind of analogues of the lactones and fluconazole was decided to test in this study. The hypothesis of this study needed to be clarified by explaining this issue.

·       The experiment on the efficacy of these new analogues is based on the in silico. On the other hand, no information is given regarding the absorption of the new analogues.

·       Conclusion should be based on the significant finding of this study. The last paragraph of the conclusion is more about the aim and expectation of this study that it should be given at the end of the introduction.

Besides these significant statements, the following minor revisions should be considered.

·       Only 3 keywords were provided. More relevant keywords would be better.

·       Half of the caption of Figure 1 is written in italics.

·       Line 3: add the before use “a factor limiting the use of azole drugs against”

·       Line 76: add the before inhibition “is based on the inhibition of the expression”

·       Line 81: add the before accumulation “This causes the accumulation of toxic 14-methyl sterols”

·       Line 82: remove the before cell “with the synthesis and growth of cell..”

·       Line 107: revise the sentence as; “more and more research has been carried out on new azole drugs against.”

·       Line 128: revise the sentence: “ROS destroys DNA deposited in the cell nucleus and degrades enzymes and...”

·       Line 181: revise the sentence as; This allowed us to obtain a less toxic compound (Fig. 10)…”

·       Line 256: revise the sentence as; better in water than fats and lactones 2-4 and vice versa.”

·       Line 274: revise the sentence as; … mammalian organisms.”

·       Line 289: revise the sentence as, “…freely in alcohol, is blocked (Fig. 4, 11).”

·       Line 293: revise the sentence as; … against the reference strains Candida albicans…”

·       Line 343: revise the sentence as; …” The synergistic effect of these two compounds was indicated by the green colour,…”

·       Line 361: revise the sentence as; “Thus, their combined effect is less than the sum of the…”

·       Line 371: revise the sentence as; “…was several dozen times lower than in case…”

·       Line 398: revise the sentence as; “The results from the experiment confirmed that in almost all cases,…”

·       Line 424: revise the sentence as; “…less active fluconazole derivatives compensate for their activity so that a mixture…”

·       Line 541: revise the sentence as; “The solution was stirred for four days at room temperature, and then the temperature…”

·       Line 571: revise the sentence as; “…was added into the solution via a syringe for 90 mins.”

·       Line 577: revise the sentence as; “…organic layer was separated, and the residual water solution was…”

·       Line 586: revise the sentence as; “The hot reaction mixture was filtered, and the solid residue was…”

·       Line 615: revise the sentence as; “…chloride (0.18 g, 0.0012 moles) was used as the substrate.”

·       Line 664: revise the sentence as; “In a 50 mL flask, 25 mL of methanol was placed,…”

Line 766: revise the sentence as; “…compounds in the culture mixture were greater than that of the compounds alone.”

Author Response

(The authors gave the same response as above.)

Reviewer 3 Report

The authors synthesized variants of phthalide lactones and fluconazoles, tested their individual and combined activities against several strains of Candida. Overall, it is a good and solid work with logical scientific writing. 

Below are the comments:

  1. Better to put Table 1 to supplementary information, and change the content into a bar plot, which is easier to read. Please also do the same change for Table 3a/b. Data is great, but it is slightly difficult to read and do comparison.

  2. Based on the activity data from compound 1 and 5, the fluoro groups probably played an important role in its bioactivity. For compound 6, 7, what is the argument to not also test the fluoro variant? Is there previous data? It would be interesting to see the variant of 6, 7 with a difluoro ring.

Author Response

(The authors gave the same response as above.)

Reviewer 4 Report

The article describes the Synergic Effect of Phthalide lactones and fluconazole and its 2 new analogs as a factor Limiting the use of azole drugs against 3 candidiassis, which is a relevant theme because Fluconazole in Therapy Can Lead to Hepatotoxicity. It is a Significant Problem for People with Reduced Immunity. Research in this area is of great importance since it would reduce the fluconazole dosage. However, I have some questions regarding the article:

The abstract will be necessary to develop better the theoretical foundation and synergism methodologies used.

Also, I have a question for the authors:

The methodology of Response Curves and Isobolograms, based on the IC50 generated by the curves, has been widely preached to demonstrate synergism, but it was not the method used to investigate synergism in the article. How would the authors explain this fact?

Author Response

(The authors gave the same response as above.)

Round 2

Reviewer 4 Report

All of the clarification and corrections were carried out by the Authors.